# Targeting Fibrosis: The Bridge That Connects Pancreatitis and Pancreatic Cancer

**DOI:** 10.3390/ijms22094970

**Published:** 2021-05-07

**Authors:** Can Huang, Juan Iovanna, Patricia Santofimia-Castaño

**Affiliations:** Centre de Recherche en Cancérologie de Marseille (CRCM), INSERM U1068, CNRS UMR 7258, Aix-Marseille Université and Institut Paoli-Calmettes, Parc Scientifique et Technologique de Luminy, 163 Avenue de Luminy, 13288 Marseille, France; can.huang@inserm.fr (C.H.); juan.iovanna@inserm.fr (J.I.)

**Keywords:** pancreatic fibrosis, pancreatitis, pancreatic cancer, NUPR1

## Abstract

Pancreatic fibrosis is caused by the excessive deposits of extracellular matrix (ECM) and collagen fibers during repeated necrosis to repair damaged pancreatic tissue. Pancreatic fibrosis is frequently present in chronic pancreatitis (CP) and pancreatic cancer (PC). Clinically, pancreatic fibrosis is a pathological feature of pancreatitis and pancreatic cancer. However, many new studies have found that pancreatic fibrosis is involved in the transformation from pancreatitis to pancreatic cancer. Thus, the role of fibrosis in the crosstalk between pancreatitis and pancreatic cancer is critical and still elusive; therefore, it deserves more attention. Here, we review the development of pancreatic fibrosis in inflammation and cancer, and we discuss the therapeutic strategies for alleviating pancreatic fibrosis. We further propose that cellular stress response might be a key driver that links fibrosis to cancer initiation and progression. Therefore, targeting stress proteins, such as nuclear protein 1 (NUPR1), could be an interesting strategy for pancreatic fibrosis and PC treatment.

## 1. Introduction

Pancreatitis is triggered by a variety of factors including the activation of pancreatic enzymes, resulting in pancreatic tissue self-digestion accompanied by pancreatic tissue edema, bleeding, and inflammatory necrosis [1,2,3]. Patients with pancreatitis suffer fever, nausea, vomiting, abdominal distension, abdominal pain, and other symptoms [4]. Pancreatitis can be divided into two types: acute pancreatitis (AP) and CP. AP is an acute pancreatic disease with strong pain, which has greater occurrence in middle-aged adults [5,6,7]. On the contrary, most CP patients develop mild disease, and the symptoms are usually not noticeable [8]. In the course of CP, patients often suffer pain and exocrine and endocrine insufficiency [9]. Most patients with AP recover completely after receiving the right treatment. Adjusting one’s diet and stopping smoking and alcohol ingestion can completely restore pancreatic homeostasis [10]. However, without the correct treatment, AP may deteriorate into CP, and the pathological changes caused by CP are often irreversible [11].

PC is a kind of gastrointestinal tumor with high malignancy, which is difficult to diagnose and treat [12]. Pancreatic ductal adenocarcinoma (PDAC) represents 90% of PC, with a poor prognosis (the 5-year survival rate of PDAC is less than 10%) [13]. Most PDAC patients have metastases after diagnosis, which cannot be treated by surgery [14]. In addition, the etiology of PDAC is complex, and both genetic and environmental factors are involved in the disease progression. Specific mutations in genes, such as tumor protein P53 (TP53), V-Ki-ras2 Kirsten rat sarcoma viral oncogene homolog (KRAS), cyclin-dependent kinase inhibitor 2A (CDKN2A), or SMAD family member 4 (SMAD4) increase the risk of developing PDAC [15]. Other significant risk factors for PDAC development that have been associated with this disease are smoking, diabetes, alcoholism, and obesity [16].

Pancreatitis and PC are two diseases of varying degrees in the pancreas, with similar symptoms. It is frequently necessary to exclude PC in the diagnosis of pancreatitis. From an imaging perspective, pancreatitis and PC are easily confused on magnetic resonance imaging (MRI) or computerized tomography (CT), as only specific imaging features allow one to discriminate between these diseases in the differential diagnosis [17]. Therefore, the screening for PC should combine biochemical tests and pathological diagnosis of tumor markers, such as carbohydrate antigen 199 (CA199), cancer antigen 125 (CA125), carcinoembryonic antigen (CEA), and carbohydrate antigen 50 (CA50) [18]. Most PC patients have a history of CP, which shows that there is a high correlation between PC and CP [19]. Although PC has high genetic factors, pancreatitis and PC have a variety of common pathogenic factors, such as long-term smoking, alcohol abuse, or high protein and high-fat diets [20,21]. In recent years, different works have demonstrated that the processes of wound healing and tumor fibrosis have strong similarities. Interestingly, pancreatic fibrosis is also one of the main pathological features of CP, suggesting a strong relationship between PC and CP [22]. However, whether CP increases the risk of PC and promotes the occurrence and development of PC through tissue fibrosis remains an open question. Thus, in this work, we discuss the role of pancreatic fibrosis in pancreatitis and PC development as well as the recent findings targeting this process (Figure 1).

## 2. The Role of Fibrosis in Pancreatitis

CP is a pancreatic fibrotic syndrome associated with genetic, environmental, and/or other risk factors [23]. Clinically, CP patients have recurrent abdominal pain, nausea, dyspepsia symptoms, and different complications including fat-soluble vitamin deficiency, exocrine dysfunction, metabolic bone disease, and diabetes [24]. The pathological features of CP include pancreatic fibrosis, acinar injury, pancreatic calcification, and exocrine and endocrine dysfunction [25]. CP is a persistent pathological response to substantial injury or stress. Irreversible fibrosis is one of the most typical pathological features of CP and deeply affects the physiological function of the pancreas [26]. Thus far, there is no clinical treatment that can reverse the inflammatory damage associated with CP. Therefore, CP treatment is focused on relieving the symptoms and screening and treating disease-related complications [27].

### 2.1. Pancreatic Fibrosis Promotes Inflammation

Numerous studies have shown that pancreatic fibrosis not only is a feature of pancreatitis disease but also has an active role in CP development [28,29,30]. For instance, alcohol consumption triggers pathological changes in the pancreas, leading to pancreatic fibrosis, causing alcoholic CP [31]. Interestingly, increased trypsinogen content in the pancreas is a pivotal event in the initiation of alcoholic CP, although the mechanism remains elusive [32,33]. Serine protease inhibitor Kazal type 1 (SPINK1) acts as a trypsin inhibitor, and its mutation dramatically increases the risk of alcoholic CP [34]. Recent evidence shows that SPINK1-associated pancreatitis or alcohol-induced CP can be characterized by progressive parenchymal fibrosis [35,36]. In addition, a variety of immune cell types, such as monocytes and macrophages, have been detected in the dense fibrotic areas in pancreatic cancer. Indeed, as part of the innate immune response, these immune cells can be recruited by inflammatory signals [37,38,39,40]. However, previous studies have shown that monocytes are recruited into damaged tissues, subsequently differentiating into macrophages, stimulating the synthesis of collagen and fibronectin (FN), and participating in the process of pancreatic fibrosis [41]. Furthermore, macrophages interact with neighboring cells in a cytokine-dependent manner to accelerate the formation of pancreatic fibrosis during pancreatitis [42,43]. Recent studies have shown that the oral administration of camostat mesilate (CM), a drug for CP treatment, reduces pancreatic fibrosis and subsequent inflammation by inhibiting the activity of monocytes [44]. In sum, macrophages, as the major infiltrating immune cells in the tumor microenvironment (TME), play an important role in pancreatic fibrosis and inflammation.

### 2.2. Pancreatic Stellate Cells (PSCs) Are Key Mediators of Fibrosis in Pancreatitis

Based on the close relationship between pancreatitis and fibrosis, some studies have shown that pro-inflammatory cytokines induce PSCs activation [45]. The activated PSCs further secrete more inflammatory factors, such as monocyte chemotactic protein 1 (MCP-1), which regulates fibrosis through its cognate CC chemokine receptor 2 (CCR2) [46]. Therefore, PSCs activation is considered to be the core event in the development of pancreatic fibrosis, suggesting that targeting PSCs is a potential strategy for CP therapy. Hydrogen peroxide-inducible clone-5 (Hic-5) is a member of the paxillin family, which acts as a molecular scaffold, and its expression leads to a poor prognosis in PC patients [47]. In caerulein-induced CP, the expression of Hic-5 was strongly up-regulated in activated PSCs in the fibrotic tissue [48]. As such, decreasing the expression of Hic-5 significantly attenuated pancreatic fibrosis and PSCs activation in experimental CP mice. Therefore, Hic-5 is an important therapeutic target to reduce pancreatic fibrosis and delay CP [48]. Vitamin deficiency is usually present in patients with pancreatitis and PC and may result from pancreatic insufficiency [49,50]. Interestingly, dietary interventions, such as long-term consumption of vitamin-rich vegetables and fruits, also slow down the CP caused by pancreatic fibrosis [27]. Vitamins C and E act as classical antioxidants, and both have shown a potent anti-fibrotic and anti-inflammatory action by preventing oxidative damage in several organs, including the pancreas [51,52,53,54]. In some studies, vitamins A, D, and K also exert a protective role in the inflammatory response, probably through their antioxidant properties, implying the importance of oxidative stress in inflammation [55,56,57]. Fibrosis is usually considered irreversible in CP, but some studies have shown that pancreatic fibrosis can be reversed in the early stage [20,26,58,59]. However, fibrosis prevention is an effective strategy to reduce CP either by drug treatment or dietary adjustments [60].

## 3. The Role of Fibrosis in PC

Currently, there is increasing evidence that PC is a chronic inflammatory disease, as with fibrosis being one of the main pathological characteristics [61]. It is well known that there is a close relationship between pancreatic fibrosis and PC [62]. Many pro-fibrotic cells and cancer-associated fibroblasts (CAFs) are abundantly present in PDAC [63]. Most studies have shown that pancreatic fibrosis level is closely related to the survival rate of the patients after chemotherapy [64,65]. Thus, having a quantitative and reproducible method, evaluating fibrosis might be more accurate than either pathologic regression grade or response evaluation criteria in solid tumors (RECIST) score [66].

### 3.1. Pancreatic Fibrosis Promotes PC Progression

Pancreatic fibrosis is a defining hallmark of PDAC occurrence and prognosis. Interestingly, a set of genes were recently reported to be involved in the development of pancreatic fibrosis and PC. For example, C-X-C motif chemokine receptor 2 (CXCR2), functionally expressed in leukocytes, such as neutrophils, natural killer cells (NK cells), monocytes, macrophages, and T cells, regulates the migration of neutrophils to inflammatory sites by binding to Interleukin-8 (IL-8) [67,68]. CXCR2 knockout mice showed higher levels of pancreatic fibrosis and increased the malignancy of PDAC in vivo, indicating that *CXCR2* played an important role in the transition from pancreatic fibrosis to PC [69]. Besides chemokine receptors, some metabolic enzymes are also involved in regulating pancreatic fibrosis. Long-chain acyl coenzyme A synthase 3 (ACSL3) is a lipid metabolizing enzyme, which is up-regulated in PC and related to the increased fibrosis. Interestingly, Sebastiano and colleagues demonstrated that *ACSL3* knockout prevents pancreatic fibrosis and delays the PDAC development in mice [70]. A disintegrin and metalloproteinase domain-containing protein 10 (ADAM10) also correlates with the occurrence and development of PC [71]. Both gene-targeting and drug-targeting ADAM10 reduced radiotherapy-induced pancreatic fibrosis and tissue tension, decreasing the migration and invasion of tumor cells, increasing the tumor sensitivity after radiation, and ultimately prolonging the survival of mice [72]. Furthermore, a recent study confirmed that pancreatic fibrosis reduces the lethality and immunity of immune cells to pancreatic tumor cells, thus promoting PDAC progression [73]. Altogether, pancreatic fibrosis is not only a marker in the formation, development, and prognosis of PC, but also has active participation in cancer disease.

### 3.2. CAFs Contributes to Drug Resistance

Currently, treatment of PC is a big challenge; thus, patients face a poor prognosis, and drug resistance is a major problem in PC therapy [74]. The tumor tissues in PC are composed of a small proportion of cancer cells. In fact, there is an extensive amount of proliferative matrix surrounding the cancer cells, that represents up to 90% of the tumor mass [75]. These proliferative matrices include ECM, CAFs, endothelial cells, and invasive immune cells [76]. These abnormally rich matrices act as a tight blockade to prevent chemotherapeutic drugs from penetrating into the tumor and playing their anti-cancer role, which is one of the important factors that endow cancer cells with chemotherapeutic resistance. Among them, CAFs are the most critical part of TME regulation.

#### 3.2.1. Therapeutic Targeting of the Crosstalk between CAFs and PC

FN assembled by CAFs is an ECM integrin-binding protein. FN promotes fiber formation, provides a track for the migration of cancer cells, and mediates the directional migration of cancer cells [77,78]. Moreover, many signaling molecules produced by CAFs directly participate in regulating nearby cancer cells, thereby stimulating proliferation, invasion, and chemical resistance, which promotes PC malignancy [79]. The consumption of matrix in PDAC blocks some signaling pathways, so it significantly improves the effect of chemotherapy [80]. In vivo studies have shown that the Hedgehog receptor Smoothened (SMO) overexpression in CAFs is an important mechanism of Hedgehog (Hh) signal transduction in pancreatic stromal cells, and the Hh signaling pathway has a close interaction with the tumor matrix [81]. N-myc downstream-regulated gene 1 (*NDRG1*) is considered to be a potential anticancer gene, and its expression correlates with the differentiation of tumors [82]. Recent studies showed that targeting NDRG1 blocks the crosstalk between PC cells and matrix [83]. The TGF-β/Smad4 signaling axis plays an important role in regulating the TME and mediating tumor-stroma crosstalk [84]. The Met/HGF pathway not only involves the interaction between cancer cells and activated PSCs but also takes part in the crosstalk between tumor and matrix [85]. The complex NT-S100A8/TGF-β1 is also involved in the crosstalk between PDAC and stromal cells in some specific PDAC cell lines [86]. In addition, many studies have confirmed that microRNAs (MIRs) related to epigenetic regulation is a key factor in the formation of the TME, because MIRs are involved in the transformation of normal fibroblasts (NFs) to CAFs, and MIRs released from CAFs affect various features of cancer cells such as tumor migration, tumor invasion, metastasis, and drug resistance [87,88]. Pasireotide (Som230) is a novel multireceptor-targeted somatostatin analog, which inhibits the secretion of symbiotic sulfate transporter 1 (SST1) in CAFs, thus eliminating the interaction between CAFs and PDAC [89]. Insulin/IGF-1R signal is also involved in the crosstalk between cancer cells and matrix, and research on compounds targeting Insulin/IGF signal in the treatment of PDAC entered into clinical trials in phase II [90].

#### 3.2.2. Targeting CAFs in Combination with Chemotherapy, a Field to Explore

Besides the heterogeneity of PDAC itself, the complex matrix crosstalk of tumor cells in the TME also endows cancer cells with resistance to anticancer drugs, which makes the current targeted therapy for some oncogenes weak [91]. Therefore, depletion of dense matrix or destruction of its crosstalk with tumor tissue overcomes the resistance of cancer cells to chemotherapy agents and enhances the anti-cancer effect.

For instance, IL-1β/IRAK4 is the feedforward signal of the tumor matrix with a very high expression level in cancer development, and disrupting the tumor-stroma IL-1β/IRAK4 feedforward circuitry improves the chemotherapy in PDAC [92]. Reducing perlecan in the matrix decreases the contact and communication between the matrix and cancer cells. Depleting perlecan in the stroma and combining with chemotherapy drugs such as gemcitabine (GEM) or Abraxane can prolong the survival rate of PDAC mice [93]. Erdafitinib, a selective pan-fibroblast growth factor receptor (FGFR) inhibitor approved by FDA, reduces the drug resistance of PC cells by targeting tumor fibroblast receptors to prevent the crosstalk between CAFs and cancer cells [94]. In clinical studies, Vismodegib, an orally bioavailable small molecule, has been found to inhibit Shh pathway and to reduce the production of stromal cells, thereby enhancing the anti-cancer effect of GEM on PC [95]. The combination of the least toxic anti-cancer drugs and anti-matrix drugs has gradually become a promising new cancer treatment [96].

#### 3.2.3. CAFs Activation Suppresses Tumor Immune Response

Recent studies showed that the matrix in the tumor stroma also participates in the immune response [97]. For example, high expression of Caveolin-1 (CAV1), a membrane-associated scaffold protein, enhances the secretion of Interleukin-6 (IL-6) and IL-8 in CAFs and promotes PC invasion, while the down-regulation of CAV1 slows down the proliferation of PC cells [98]. Netrin-G1 (NETG1), a lipid anchored protein promotes CAFs to secrete glutamine, glutamate, and cytokines through p38/Fra-1 and Akt/4E-BP1 pathways, thus supporting the survival of PDAC cells under low nutritional conditions and reducing the antitumoral effect of NK cells on PDAC cells [99]. Hypoxia-induced fibrosis can inhibit the infiltration of T cells into the tumor, and the continuous activation of hypoxia-inducible factor 1 alpha (HIF-1α) can negatively regulate the signal transduction function of T cell receptors [100]. In human pancreatic fibrosis, macrophages are closely linked to PSCs, which may promote the activation of CAFs during chronic pancreatitis [43]. Tumor-associated macrophages promote cancer fibrosis by regulating ECM [101]. Monocytes can be recruited by CAFs via the IL-8/CXCR2 pathway and differentiate into macrophages that support tumor growth [102]. Therefore, CAFs can not only directly contact PC cells and secrete metabolites but also participate in the immune regulation of the TME. The strategy of targeting the interaction between CAFs, the immune system, and cancer cells can enhance the anti-tumor activity. Recently, it has been reported that knockout of the adhesion molecule, *cadherin 11* (*CDH11*), which is mainly produced by CAFs, inhibits the growth of the pancreatic tumor, increases the response to GEM, reverses the immunosuppression of CAFs, and ultimately significantly prolongs the survival of mice [63].

#### 3.2.4. CAFs Heterogeneity Is a Challenge in Cancer Therapy

Several subtypes of CAFs have been identified, including myofibroblastic CAFs (myCAFs), inflammatory CAFs (iCAFs), and antigen-presenting CAFs (apCAFs) [103]. MyCAFs with a high expression of actin alpha 2 (ACTA2) were first identified in PC. For a long time, MyCAFs were considered to be the only CAF population, because α-SMA is widely expressed in CAFs [104,105]. MyCAFs play a major role in regulating the deposition and remodeling of ECM, highlighting the important role of myCAFs in promoting pancreatic fibrosis and pancreatitis [106]. ICAF is a subtype of CAFs with a high expression of Ly6C. ICAFs are driven by tumor secretory factors, such as Interleukin-1 alpha (IL-1α) and Interleukin 1 beta (IL-1β), and gather at the edge of the tumor [107]. IL-1α signaling also drives the autocrine signaling in iCAFs, which helps to maintain the inflammatory phenotype. ICAFs produce a variety of cytokines and chemokines (CCL2, CCL7, IL-6, and CXCL12) and may have a stronger tumor-promoting effect than myCAFs [108,109]. ICAF stimulates the proliferation and angiogenesis of PC cells and promotes PC development [110]. Furthermore, iCAFs inhibit the immune response through recruit regulatory T cells (Tregs) and myeloid-derived suppressor cells (MDSCs) [111]. ApCAFs are the CAFs with antigen-presenting function, and the expression of major histocompatibility II (MHC II) is the major feature of apCAFs [110]. ApCAFs present antigens to T cells and affect T cell immunity [110]. ApCAFs can induce CD4+ Tregs differentiation through antigen-dependent T cell antigen receptor (TCR) ligation, reduce the anti-tumor immune response by changing the ratio of CD8+ T cells to Tregs and prevent PC cells from being monitored by immune cells [110,112,113]. According to the latest research, all three of these CAFs can transform into each other, which emphasizes the dynamic process of TME [114,115].

The majority of studies suggested that CAFs promote the development of PDAC. Recently, some unexpected results showed that myCAFs depletion may also promote PDAC development and metastasis [116]. It possibly depends on whether these CAFs are invasive (carcinogenic) or tumor suppressor CAFs because they might play different or even opposite roles in PDAC development [117]. Considering these studies, it would be necessary to distinguish the subtypes of CAFs, identify different markers, and explore the reasons for the high CAFs heterogeneity.

## 4. Cellular Stress Response Led to Pancreatic Fibrosis

When healthy cells suffer constant damage such as genotoxicity, protein, or lipid damage, cellular stress response confers the cellular adaptation that prevents cell death and promotes the transformation of healthy cells into tumor cells. Previous studies have clearly shown that stress proteins play an important role in maintaining the homeostatic microenvironment in both healthy and tumor cells [118,119,120]. Moreover, oxidative stress is required for driving metabolic reprogramming and the re-establishment of antioxidant systems in cancer cells [121]. Therefore, it is necessary to address how to target pancreatic fibrosis by reducing reactive oxidative species (ROS) or targeting important stress proteins.

### 4.1. ROS Scavengers for Treating Pancreatic Fibrosis

A large number of studies support oxidative stress as a triggering factor for pancreatic fibrosis. Oxidative stress directly promotes the activation of quiescent PSCs and the formation of an extensive amount of ECM, and subsequently promotes excessive fibrosis [122,123]. Meanwhile, oxidative stress also aggravates the damage of pancreatic cells in pancreatitis [124]. During the inflammatory phase, CAFs are recruited and activated under oxidative stress, which induces changes in the morphology and the functions of CAFs. However, this activated phenotype was prevented by several antioxidants [125]. For example, the ROS induced by H_2_O_2_ promotes the activation of PSCs, while resveratrol prevents the activation of PSCs by reducing the production of ROS [126]. Moreover, melatonin, at pharmacological concentrations, has shown a concentration-dependent decrease in cell viability in rat [127] and human [128] PSCs by modifying the redox state of the cells. Dimethyl fumarate (DMF) promotes the activation of nuclear factor erythroid 2-related factor 2 (NRF2) and the expression of downstream antioxidant genes, eliminating intracellular ROS, inhibiting the activation of PSCs, and reducing the pancreatic fibrosis level [129]. ROS-induced inflammation caused pancreatic cell death through RF2/NF-kB and SAPK/JNK pathways. The antioxidant N-acetyl cysteine (NAC) rescues cell viability by decreasing oxidative stress and inflammation in primary pancreatic cells [130]. Diethyldithiocarbamate is a kind of superoxide dismutase (SOD) inhibitor, was able to induce pancreatic fibrosis by increasing ROS in treated rats [131]. Vitamin E reduces oxidative stress and collagen deposition during CP, thereby reducing pancreatic fibrosis in cerulein-treated mice [124]. Theobromine and scoparone reduce the oxidative stress of pancreatic cells, inhibiting the activation of PSCs and attenuating pancreatic fibrosis through the TGF-β/Smad signaling pathway [132]. In mice, long-term administration of antioxidants prevents PSCs activation (by high glucose-diet) and subsequent fibrosis cascade. Coenzyme Q10 (CoQ10) reduces oxidative stress response, blocks ROS-induced PI3K/Akt/mTOR signaling pathway, decreases pancreatic fibrosis, and prevents the activation of PSCs [133]. Therefore, CoQ10 may be a drug candidate to treat pancreatic fibrosis [134].

Clinical research data show that antioxidant supplementation reduces the level of oxidative stress in patients with idiopathic CP and alcoholic pancreatitis and then weakens the process of pancreatic fibrosis [135]. More evidence supports that oxidative stress has an essential role in the progression of pancreatic fibrosis. Compared with other human organs, such as the liver or the kidney, the pancreas is more sensitive to long-term oxidative stress, triggering inflammation [123]. It has been suggested that ROS-induced oxidative stress can cause persistent damage to the biomacromolecules, such as DNA, RNA and proteins in pancreatic cells, promoting metabolic reprogramming and antioxidant system remodeling. Therefore, the use of antioxidants can reduce oxidative stress response, inhibit pancreatic fibrosis, and reduce the transformation of cells, compromising tumor development.

### 4.2. Targeting Stress-Inducible Protein NUPR1 for Treating Pancreatic Fibrosis and PC

NUPR1 is a stress-induced protein, which is over-activated in the damaged pancreas cells in AP and CP and plays an important role in PC development [136,137,138]. In addition, NUPR1 plays a crucial role in the fibrosis of multiple organs and tissues. For example, NUPR1 activated in the fibroblasts and the renal tubular epithelial cells promotes renal interstitial fibrosis [139]. Similarly, type I collagen and FN promote glioma progression by the activation of NUPR1 [140]. A recent study also found that knockout of *NUPR1* decreases cardiac fibrosis and partially restores cardiac function [141]. In a spontaneous mouse model of CP, the oral protease inhibitor CM inhibits CP and pancreatic fibrosis by reducing the expression of NUPR1 [142]. Therefore, we proposed that NUPR1 plays an indispensable role in the progression of fibrosis, and inactivation of NUPR1 is a promising strategy for preventing fibrosis.

In this line, our recent studies have shown that ZZW-115, a powerful inhibitor of NUPR1, is able to kill cancer cells from different tumors, including PC. ZZW-115 is extremely effective in every subtype of PC, but also enhances the sensitivity of cancer cells to chemotherapeutic drugs [143,144]. Importantly, ZZW-115 cannot improve the sensitivity of the untransformed fibroblasts to chemotherapeutic drugs [145]. Interestingly, NUPR1 as a transcriptional factor is not only activated in oxidative stress but also activated in response to other cellular stress, such as endoplasmic reticulum stress (ER stress) and metabolic stress [146,147,148]. Our recent research shows that ZZW-115 treatment triggers ROS production, thus highlighting the role of NUPR1 in the oxidative stress response [143]. In conclusion, NUPR1 inhibitors have a variety of interesting effects in the treatment of PC, including attenuating fibrosis to slow the PC progression, killing PC cells directly through a variety of ways of death, and fighting drug resistance of cancer cells [149].

## 5. Conclusions

CP is characterized by persistent and permanent damage in pancreatic tissue [23]. The endocrine and exocrine compartments of the damaged pancreas are gradually lost and replaced by atrophy or fibrosis [23]. The CP development leads to organ dysfunction and increases the risk of PC development. Pancreatic fibrosis is also a typical feature of PC, which promotes the recruitment and activation of CAFs [150]. Pancreatic fibrosis can be used as a diagnostic marker of PC. Besides the traditional imaging methods, evaluating the level of pancreatic fibrosis could improve the diagnosis. Moreover, detecting inflammatory and oxidative stress indicators will contribute in the future to a better understanding of PC development, but also the diagnosis, prevention, and prognosis of the disease [151]. Furthermore, the development and application of the new generation of histology needles provide the possibility to analyze the TME of PC via endoscopic ultrasound [152,153,154], collecting the tumor tissue and allowing the analysis of the tumor matrix.

In addition, pancreatic fibrosis leads to hypoxia in the pancreatic tumor, which causes more oxidative stress response, promoting tumor aggressiveness, increasing drug resistance in cancer cells, and thereby causing higher patient mortality rates [155]. Interestingly, NUPR1, a stress protein activated in pancreatitis, promotes fibrosis, inflammation, and cancer initiation and development, indicating that NUPR1 is essential for TME. Collectively, cellular stress response drives fibrosis, playing a vital role in the transformation from CP to PC. Therefore, prevent fibrosis or targeting stress proteins, such as NUPR1, could be a promising therapeutic strategy for PC and CP therapy.

## Figures and Tables

**Figure 1 ijms-22-04970-f001:**
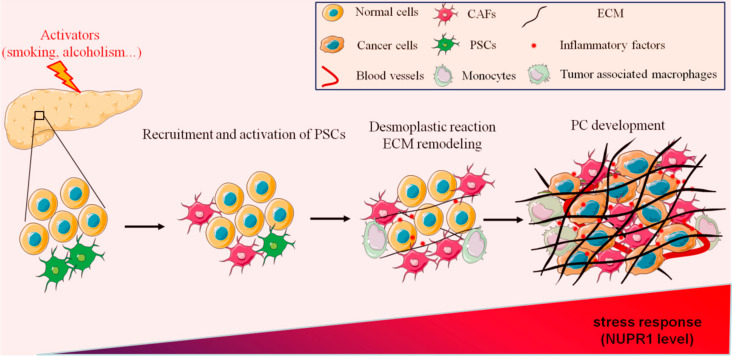
Schematic representation of the mechanisms involved in PC initiation.

## Data Availability

No new data were created or analyzed in this study. Data sharing is not applicable to this article.

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
