# Peer review of "Targeting Fibrosis: The Bridge That Connects Pancreatitis and Pancreatic Cancer"

_ijms, 2021, doi:10.3390/ijms22094970_

Round 1

Reviewer 1 Report

This is a comprehensive review on pancreatic fibrosis and its role in pancreatic cancer development and treatment.

From a clinical point of view, I found this paper interesting and useful to better understand biochemical and molecular mechanisms. I would like the Authors to add a point in the discussion section. All the information you included in your study comes from surgical specimens. However, about 80% of patients with pancreatic cancer will never undergo resection. Thus, the only histologic material we can use from these patients for personalized medicine will come from a biopsy performed during endoscopic ultrasound. This point is important because, recently, new generation histology needles has become available, demonstrating higher histologic yield compared with both standard FNA needles (see Oppong KW, et al. Endoscopy. 2020 Jun;52(6):454-461. doi: 10.1055/a-1114-5903) and old generation histology needles (see Crinò SF, et al. Gastrointest Endosc. 2020 Sep;92(3):648-658.e2. doi: 10.1016/j.gie.2020.05.016). Moreover, these needles demonstrated to be able to collect adequate material for molecular profiling (see Kandel P, et al. Endoscopy. 2021 Apr;53(4):376-382.  doi: 10.1055/a-1223-2171. Epub 2020 Aug 6. PMID: 32767288.). Further studies are needed to investigate the possibility to perform specific analyses on tumor matrices and on microenvironment on tissue collected via endoscopic ultrasound to use the information you clearly reported in your paper on the majority of patients with pancreatic cancer.

Author Response

From a clinical point of view, I found this paper interesting and useful to better understand biochemical and molecular mechanisms.

Thanks to the reviewer for these words

I would like the Authors to add a point in the discussion section. All the information you included in your study comes from surgical specimens. However, about 80% of patients with pancreatic cancer will never undergo resection. Thus, the only histologic material we can use from these patients for personalized medicine will come from a biopsy performed during endoscopic ultrasound. This point is important because, recently, new generation histology needles has become available, demonstrating higher histologic yield compared with both standard FNA needles (see Oppong KW, et al. Endoscopy. 2020 Jun;52(6):454-461. doi: 10.1055/a-1114-5903) and old generation histology needles (see Crinò SF, et al. Gastrointest Endosc. 2020 Sep;92(3):648-658.e2. doi: 10.1016/j.gie.2020.05.016). Moreover, these needles demonstrated to be able to collect adequate material for molecular profiling (see Kandel P, et al. Endoscopy. 2021 Apr;53(4):376-382.  doi: 10.1055/a-1223-2171. Epub 2020 Aug 6. PMID: 32767288.). Further studies are needed to investigate the possibility to perform specific analyses on tumor matrices and on microenvironment on tissue collected via endoscopic ultrasound to use the information you clearly reported in your paper on the majority of patients with pancreatic cancer.

We appreciate this suggestion, and the text has been modified, including now in the discussion this important information.

Reviewer 2 Report

In the current review article, the authors Huang et al. summarize the link between pancreatic fibrosis and pancreatic cancer. The topic is highly interesting. The authors discuss some topics in detail (for example, cellular stress response), but other parts of the manuscript are unfortunately too superficial. I would recommend therefore as follow:

Line 40: the most recent cancer statistics shows that the 5-year survival rate of pancreatic cancer is 10%. In the cited publication of the current article [13], Adamska et al. mentioned that the five-year survival rate remains just around 5-7%. Adamska et al. cited Vincent et al. 2011 for this information. These are all relevant publications, but I would recommend updating the statistical data for the current manuscript. Siegel et al. Cancer Statistics, 2021. PMID: 33433946.

Line 86, 87, 95: please provide short descriptions for SPINK1, CFTR, Hic-5, with references.

Section 2.1.: the section is too superficial. Which immune cell types are involved?

In figure 1, the authors mention monocytes and tumor associated macrophages. However, there is no discussion, description about monocytes and tumor associated macrophages in the manuscript.

Line 91: the authors wrote “.. pro-inflammatory cytokines induce PSCs activation. The activated PSCs further secrete more inflammatory factors, such as MCP-1, which promotes pancreatic fibrosis”. Activated PSCs produce ECMs and support fibrosis. How MCP-1 promotes pancreatic fibrosis?

Line 100: what would be potential background mechanisms of slowing down the chronic pancreatitis by long-term consumption of vitamin-rich vegetables and fruits? How about vitamin A, vitamin D?

Line 101: are you sure fibrosis is not reversible?

Line 118: the authors mention Cxcr2, Acsl3, then ADAM10. The authors need to discuss to make the section more structured. Please provide short descriptions for Cxcr2, Acsl3, ADAM10. Which cells express Cxcr2? Ligands? Why the authors start to mention about chemokine receptor and then jumped to metabolic enzyme?

Line 151: Please provide short description for NDRG1.

Line 188: Please provide short description for CAV1.

Line 190: Please provide short description for NETG.

Line 202: myo-fibroblastic CAFs

Line 203: The authors start to mention CAF heterogeneity, then “suggested CAFs promote the development of PDAC”. Please discuss potential contribution of myCAF, iCAF, and apCAP (antigen presenting CAFs) in pancreatic cancer.

Line 204: please rephrase the sentence “some unexpected results….”. What does “the complete consumption of CAFs” mean?

Throughout the manuscript, I found only in line 193 “reducing the killing effect of NK cells on PDAC cells” as specific description regarding immune cells. How about macrophages? How about T cells?

The authors discuss about the role of ROS and NUPR1 in pancreatic fibrosis. This part is interesting. Weis et al. has shown that P8 deficientcy increases cellular ROS (Weis et al. Arch Biochem Biophys.2015) PMID: 25475530. The role of p8/NUPR1 in ROS can be different between cell types?

Please check throughout the manuscript, abbreviations should be stated in full acronym before being abbreviated.

Author Response

In the current review article, the authors Huang et al. summarize the link between pancreatic fibrosis and pancreatic cancer. The topic is highly interesting.

We thanks the reviewer for these words

The authors discuss some topics in detail (for example, cellular stress response), but other parts of the manuscript are unfortunately too superficial. I would recommend therefore as follow:

Line 40: the most recent cancer statistics shows that the 5-year survival rate of pancreatic cancer is 10%. In the cited publication of the current article [13], Adamska et al. mentioned that the five-year survival rate remains just around 5-7%. Adamska et al. cited Vincent et al. 2011 for this information. These are all relevant publications, but I would recommend updating the statistical data for the current manuscript. Siegel et al. Cancer Statistics, 2021. PMID: 33433946.

We thank the reviewer for this comment. We update the reference.

Line 86, 87, 95: please provide short descriptions for SPINK1, CFTR, Hic-5, with references.

We appreciate this suggestion, and the text has been modified.

Section 2.1.: the section is too superficial. Which immune cell types are involved?

Thanks for your suggestion, and we have included more information regarding the immune cells and their role in the regulation of fibrosis during pancreatitis.

In figure 1, the authors mention monocytes and tumor associated macrophages. However, there is no discussion, description about monocytes and tumor associated macrophages in the manuscript.

We thank the reviewer for this comment. We now add the discussion and description about monocytes and tumor-associated macrophages in the manuscript, such as in section 3.2.3.

Line 91: the authors wrote “.. pro-inflammatory cytokines induce PSCs activation. The activated PSCs further secrete more inflammatory factors, such as MCP-1, which promotes pancreatic fibrosis”. Activated PSCs produce ECMs and support fibrosis. How MCP-1 promotes pancreatic fibrosis?

We thank the reviewer for this comment, and we have modified the text including this information.

Line 100: what would be potential background mechanisms of slowing down the chronic pancreatitis by long-term consumption of vitamin-rich vegetables and fruits? How about vitamin A, vitamin D?

Thanks for your suggestion. We have rewritten this section and discuss the potential role of anti-oxidative stress of several vitamins on inflammation.

Line 101: are you sure fibrosis is not reversible?

We agree with this suggestion, and we have included new references addressing the reversibility of fibrosis in the early stages.

Line 118: the authors mention Cxcr2, Acsl3, then ADAM10. The authors need to discuss to make the section more structured. Please provide short descriptions for Cxcr2, Acsl3, ADAM10. Which cells express Cxcr2? Ligands? Why the authors start to mention about chemokine receptor and then jumped to metabolic enzyme?

Thanks for your suggestion, we have reorganized the structure of section 3.1, providing more information about these genes.

Line 151: Please provide short description for NDRG1.

We thank the reviewer for this comment, and we have modified the text according to the suggestion.

Line 188: Please provide short description for CAV1.

We thank the reviewer for this comment, and we have modified the text according to the suggestion

Line 190: Please provide short description for NETG.

We thank the reviewer for this comment, and we have modified the text according to the suggestion

Line 202: myo-fibroblastic CAFs

Thanks for your suggestion, we included the description of myCAFs in section 3.2.4.

Line 203: The authors start to mention CAF heterogeneity, then “suggested CAFs promote the development of PDAC”. Please discuss potential contribution of myCAF, iCAF, and apCAP (antigen presenting CAFs) in pancreatic cancer.

Thanks for your suggestion, we discuss the role of the different CAFs, in section 3.2.4.

Line 204: please rephrase the sentence “some unexpected results….”. What does “the complete consumption of CAFs” mean?

Thanks for your suggestion. It means “MyCAFs depletion”, we now correct it.

Throughout the manuscript, I found only in line 193 “reducing the killing effect of NK cells on PDAC cells” as specific description regarding immune cells. How about macrophages? How about T cells?

Thanks for your suggestion. In section 3.2.3, we discuss the roles of macrophages, T cells, and monocytes

The authors discuss about the role of ROS and NUPR1 in pancreatic fibrosis. This part is interesting. Weis et al. has shown that P8 deficientcy increases cellular ROS (Weis et al. Arch Biochem Biophys.2015) PMID: 25475530. The role of p8/NUPR1 in ROS can be different between cell types?

Thanks for your suggestion. In section 4.2, we discuss the role of NURP1 in oxidative stress and ROS formation.

Please check throughout the manuscript, abbreviations should be stated in full acronym before being abbreviated.

We appreciate this suggestion, and the text has been modified.

Reviewer 3 Report

Huang et al., have review “Pancreatic fibrosis: the bridge that connects pancreatitis and pancreatic cancer”. The authors have attempted to describe the role of fibrosis and cancer-associated fibroblasts in pancreatic cancers. The authors also discussed reactive oxygen species' role in the induction of PC and proposed some therapeutic strategies.  

First of the manuscript deserves extensive English editing.

Further, the content should be in-depth, as the title is broad. Please refer to this, https://doi.org/10.1016/j.apsb.2019.11.008

Instead of a broad title, the authors should have selected the particular signalling pathway, which is more predominant in the induction of fibrosis to cancer.

At the present format, the manuscript is premature for publication in this journal.

Author Response

Huang et al., have review “Pancreatic fibrosis: the bridge that connects pancreatitis and pancreatic cancer”. The authors have attempted to describe the role of fibrosis and cancer-associated fibroblasts in pancreatic cancers. The authors also discussed reactive oxygen species' role in the induction of PC and proposed some therapeutic strategies.  

First of the manuscript deserves extensive English editing.

Thanks for your suggestions. The text has been corrected.

Further, the content should be in-depth, as the title is broad. Please refer to this, https://doi.org/10.1016/j.apsb.2019.11.008

Thanks for your suggestion. This reference has been included.

Instead of a broad title, the authors should have selected the particular signalling pathway, which is more predominant in the induction of fibrosis to cancer.

We have modified the title of the manuscript.

At the present format, the manuscript is premature for publication in this journal

We thank the reviewer for this comment, we think the manuscript has improved with all the valuables comments from the reviewer's panel.

Round 2

Reviewer 2 Report

The authors have done an excellent job addressing all comments. The manuscript is now well structured and more informative. The quality of the article has significantly improved.

Author Response

         The authors have done an excellent job addressing all comments. The manuscript is now well structured and more informative. The quality of the article has significantly improved.

 We thank the reviewer for these words.

Reviewer 3 Report

Huang et al., revised the manuscript extensively by considering the reviewer(s) suggestions.

The present manuscript is more notable. Particularly, a brief discussion about innate immune responses is well placed. However, few corrections are needed before accepting the manuscript for publication.

Line 27: Pancreatitis occurs…..in the pancreas. This sentence alone is wrong. Pancreatitis can’t happen when enzymes are activated. Therefore this sentence should be connected with the next sentence.

Line 88: It is a ……. or stress. This sentence should come after the “pathological features (line 93)” sentence.

Line 23: TME should be expanded. Delete the same in line 223

Line 168: CAFs should be expanded. Delete the same in line 218

Line 171-173: Sentence is not in a flow. This may be useful “Having a quantitative and reproducible method, evaluating fibrosis might be more accurate…

Line 183: neutrophils must be included in the leukocytes; as we are discussing neutrophils migration in the connecting sentence. Please check.

Line 266: Delete the word “in vivo”; chemotherapy refers to in vivo treatment.

Line 285: Expand “FGFR”

Line 346: Which type of IL-1

Line 356: CD8+ ------- CD8+ T

Line 355-357: How ApCAFs are changing the ratio of CD8+ to Tregs. Are they directly presenting the antigens to Tregs instead of CD8+ T cells?

Author Response

The present manuscript is more notable. Particularly, a brief discussion about innate immune responses is well placed. However, few corrections are needed before accepting the manuscript for publication.

We thank the reviewer for these words.

Line 27: Pancreatitis occurs…..in the pancreas. This sentence alone is wrong. Pancreatitis can’t happen when enzymes are activated. Therefore this sentence should be connected with the next sentence.

Thanks for your suggestion. We have combined the two sentences.

Line 88: It is a ……. or stress. This sentence should come after the “pathological features (line 93)” sentence.

We appreciate this suggestion, and the text has been modified.

Line 23: TME should be expanded. Delete the same in line 223

We thank the reviewer for this comment. We updated it.

Line 168: CAFs should be expanded. Delete the same in line 218

Thanks for your suggestion. we corrected it in the text.

Line 171-173: Sentence is not in a flow. This may be useful “Having a quantitative and reproducible method, evaluating fibrosis might be more accurate…

Thanks for your suggestion, we now corrected it.

Line 183: neutrophils must be included in the leukocytes; as we are discussing neutrophils migration in the connecting sentence. Please check.

Thanks for your suggestion, we now corrected it.

Line 266: Delete the word “in vivo”; chemotherapy refers to in vivo treatment.

We thank the reviewer for this comment, and we have modified the text

Line 285: Expand “FGFR”

 Thanks for your suggestion, we have modified the text

Line 346: Which type of IL-1

We agree with this suggestion, and we have included this information

Line 356: CD8+ ------- CD8+ T

Thanks for your suggestion, we have modified the text accordingly

Line 355-357: How ApCAFs are changing the ratio of CD8+ to Tregs. Are they directly presenting the antigens to Tregs instead of CD8+ T cells?

We thank the reviewer for this comment. We have modified the text including this information.